# Confidence-Calibrated Clinical Decision Support System for Reliable Respiratory Disease Screening

Micky C. Nnamdi, J. Ben Tamo, Wenqi Shi, Vivek Chundru, Benoit Marteau, Oankar Patil, May D. Wang
*Georgia Institute of Technology*, Atlanta, USA
{mnnamdi3,jtamo3,wqshi,vchundru3,bmarteau3,opatil31,maywang}@gatech.edu

*Abstract*—With the growing adoption of computer-aided diagnostic and treatment recommendation systems in healthcare, it is essential to ensure both the accuracy and reliability of AI-enabled clinical decision support systems. In this study, we comprehensively examine existing model confidence calibration methods and propose an ensemble-based calibration approach for reliable predictions in clinical decision support systems (CDSSs). Specifically, we introduce an ENsemble-based Confidence-caLibrated deep neural network, `ENCL-DNN`, to improve respiratory disease screening using cough sounds. We also leverage local interpretable model-agnostic explanations to monitor the behavior of the CDSS, identifying the key features that contribute to its predictions and ensuring transparency in the diagnosis. By employing the ensemble-based calibration method, `ENCL-DNN` demonstrates superior performance on two publicly available respiratory audio datasets, Coswara and Cambridge, as evidenced by a 50% and a 28.74% reduction in Expected Calibration Error (ECE), respectively, compared to the uncalibrated baselines. Our experiments highlight the significance of well-calibrated deep neural networks in respiratory disease screening and the enhancement of reliability in mobile healthcare systems. By providing reliable and transparent predictions, `ENCL-DNN` has the potential to promote the wide adoption of AI-driven CDSSs and thereby improve patient outcomes through early diagnosis and intervention.

*Index Terms*—respiratory disease, clinical decision support, audio signal analysis, confidence calibration

## I. INTRODUCTION

Despite the emergent advancements in artificial intelligence (AI), very few AI-enabled clinical decision support systems (CDSSs) have been adopted in real-world clinical research or practice [1]–[3]. The increasing reliance on these probabilistic machine learning models raises critical concerns regarding their accuracy and reliability [4]–[10]. Specifically, the uncritical adoption of generated probabilities can lead to inappropriate recommendations with potentially severe consequences [11], [12]. In healthcare, the stakes are incredibly high; a miscalibrated or overconfident model can result in erroneous diagnoses, inappropriate treatments, and, ultimately, harm to patient outcomes [5], [13]. Therefore, the probabilities generated by these models should be not only accurate but also communicated with a clear understanding of their inherent uncertainties.

Confidence calibration in modern neural networks refers to the process of ensuring that the predicted probabilities (i.e., confidence) of a model accurately reflect its performance and reliability [5], [14], [15]. In other words, a well-calibrated model should produce predicted probabilities that align with

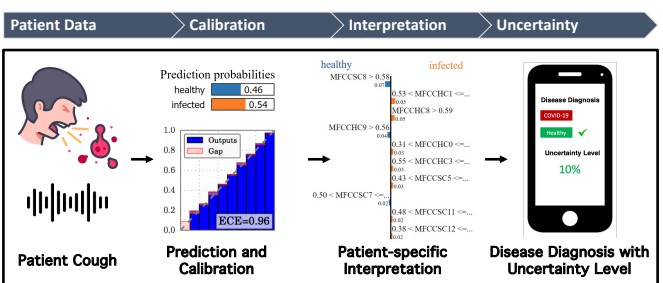

Fig. 1. Overview of the proposed clinical decision support system, `ENCL-DNN`. We first develop a deep neural network to extract hidden features from patients' cough samples for COVID-19 early diagnosis. We then calibrate model confidence to enhance the reliability of the prediction and quantify the uncertainty for a robust diagnosis. Specifically, we also leverage explainable AI to identify potentially important features from audio samples for transparent clinical decision-making.

the true likelihood of correctness. The need for confidence calibration arises because modern neural networks, especially deep learning models, tend to be overconfident in their predictions due to model depth [5].

Prior studies have introduced a variety of calibration techniques, encompassing strategies such as post-processing calibration, data augmentation for enhanced training data representation, and both Bayesian and non-Bayesian approaches to deep neural networks (DNN) for more accurate model parameter representation [5], [14], [16]–[18]. For instance, Rajaraman et al. [19] proposed calibrating deep learning models to improve the performance of medical imaging classification in the presence of class imbalance. Furthermore, Lakshminarayanan et al. [20] suggested a non-Bayesian method that involves training multiple neural networks from varied random starting points. Similarly, Krishnan et al. [11] emphasized the significance of possessing a well-calibrated model - one that not only delivers accurate predictions when confident but also indicates substantial uncertainty when its predictions are likely to be inaccurate. Despite the significant advancements in theoretical studies, there have been limited practical applications, particularly within the healthcare sector.

Uncertainty estimation is another crucial technique for assessing the reliability of model predictions [5]. Uncertainty can be broadly categorized into two main groups: aleatoric and epistemic uncertainty [21]. Aleatoric uncertainty refers to the innate unpredictability inherent in a given problem or experimental context, an aspect that remains unchanged regardless of additional empirical knowledge. Epistemic uncertainty refers

to model uncertainty that stems from gaps in knowledge or information. This kind of uncertainty is particularly prevalent in complex tasks such as medical diagnosis, where the incomplete understanding of certain symptoms or health conditions contributes to the uncertainty. In this study, we investigate the presence of epistemic uncertainty, which is closely linked to model parameters, in order to enhance the understanding of input data by calibrating model confidence.

To address these challenges, we develop an ENsemble-based Confidence-caLibrated deep neural network, `ENCL-DNN`, mitigating the gap between model capability and confidence to enhance the reliability of COVID-19 early diagnosis using audio samples. In addition, we also quantify the epistemic uncertainty during model training for a robust prediction in respiratory disease screening. Given the increasing demand for reliable and well-calibrated CDSSs, our primary motivation is to create a model that is confident in its accurate predictions and transparent in its uncertainty when predictions are less certain, thus ensuring trustworthiness and effective communication of uncertainty levels in AI-enabled CDSS outputs.

## II. METHODOLOGY

### A. Data Preprocessing

Data preprocessing consists of three stages, including (1) silence removal, (2) feature extraction, and (3) data augmentation. Firstly, we split the cough sounds into segments, divided the segments into chunks based on points of silence, and retained a small buffer of silence at the beginning and end of each chunk. The non-silent audio chunks are concatenated to form a continuous audio segment, excluding the detected silent periods. Secondly, we used the Mel-Frequency Cepstral Coefficients (MFCC) feature extraction technique to extract relevant features for COVID-19 classification. We extracted 13 MFCC features from each shallow and heavy cough sound in the Coswara dataset, and 26 MFCC features from each cough sound in the Cambridge dataset. The training data was oversampled using the SMOTE [22] technique to address the class imbalance by balancing the proportion of positive COVID-19 to healthy patients. SMOTE uses the $k$ nearest neighbors of a given sample $x_i$ in the feature space to generate new samples:

$$x_i' = x_i + \lambda(x_j - x_i), \tag{1}$$

where $\lambda$ represents a random value within the range $[0, 1]$, and $x_i'$ denotes the newly synthesized samples derived from the original sample $x_i$ and a randomly selected sample $x_j$ from the $k$ nearest neighbors of $x_i$.

### B. Model Architecture

For the classification of COVID-19 and non-COVID-19 patients, `ENCL-DNN` is composed of four densely connected layers for learning the non-linear combinations of the input features, with each node in these layers being connected to every node in the preceding and following layers. For the first three dense layers in `ENCL-DNN`, the rectified linear unit

(ReLU) activation function is employed. To mitigate overfitting, we include three dropout layers to enhance the model's capability to learn robust features. Specifically, the final dense layer adopts the sigmoid activation function, enabling the model's output to be a probability score between 0 and 1. To ensure that these probability scores are well-calibrated and reflect the true likelihood of the predicted outcomes, we apply a post-processing calibration step `ENCL`. `ENCL` ensures that these probability outputs better match the actual observed frequencies.

### C. Confidence Calibration

Calibrating DNNs is essential to ensure that the predicted probabilities accurately reflect the true likelihood of outcomes, thereby enhancing the model's ability to make reliable predictions. This process involves aligning the model's predicted confidence levels, $\hat{P}$, with their true probabilities of correctness. For instance, when a model predicts a class with confidence of 0.7, ideally, 70% of such predictions should be correct if the model is perfectly calibrated [5]. This alignment is fundamental as it bolsters the model's predictive accuracy, pinpoints uncertainties or risks, and offers more dependable insights into complex data. The calibration process includes scrutinizing and adjusting factors like algorithms, input data, and model structure. These adjustments are made using statistical methods and analytical tools to ensure that the model's outputs more accurately reflect the actual observed data.

*1) Preliminaries: Existing Confidence Calibration Methods:* We comprehensively examine the following calibration methods in this study:

- *Spline Calibration*: The spline calibration method, as described by Lucena et al. [23], employs a natural cubic spline for fitting predicted probabilities. This technique utilizes a series of knots to establish the spline's basis and to formulate the optimal calibration function, ensuring it is smooth and effective. The calibrated probability of a positive class given an input $x$ is expressed mathematically as:

$$P(y = 1|x) = \sum_{j=1}^{K} \beta_j B_j(x), \tag{2}$$

where $P(y = 1|x)$ represents the calibrated probability of the positive class given the input $x$ and $K$ denotes the number of basis functions or knots used in the spline calibration. $\beta_j$ denotes the coefficients associated with each basis function, and $B_j(x)$ represents the value of the $j$-th basis function evaluated at the input $x$. Spline calibration is particularly advantageous for its flexibility, surpassing the capabilities of piecewise constant or sigmoid functions.

- *Platt Calibration*: Platt calibration [24] assumes that the relationship between the predicted probabilities $(\alpha)$ and the true probabilities $P(y = 1|x)$ follows a sigmoidal

curve. The model has two adjustable parameters, $\alpha$ and $\beta$ and is defined by the mathematical expression:

$$P(y = 1|x) = \frac{1}{1 + \exp(\alpha f(x) + \beta)}, \quad (3)$$

where $\alpha$ and $\beta$ are real valued. Platt calibration's main drawback is that it has a limited range of possible functions. In other words, this technique can only provide accurate probability estimates if there is a logistic relationship between the output of the binary classifier and the actual probability of the positive class.

- *Beta Calibration*: Beta calibration methods leverage a richer class of calibration maps based on the beta distribution [25]. It can be defined as:

$$P(y = 1|x) = \left(1 + \frac{1}{\exp(\gamma)\frac{f(x)^\alpha}{(1-f(x))^\beta}}\right)^{-1}. \quad (4)$$

In this case, parameters $\gamma$, $\beta$, and $\alpha$ are real values that are determined when fitting the curve. Beta calibration is a flexible method, but it is still parametric. Therefore, it may not be able to correct all types of distortions, especially if the original model has non-monotonic distortions or is poorly calibrated.

- *Isotonic Calibration*: Isotonic calibration [26] fits a piecewise constant function to the classifier's score that is monotonically increasing and minimizes the mean squared error. The probability of a positive class given an input $x$ is calculated as:

$$P(y = 1|x) = s_i, \quad (5)$$

where $s_i$ is the score given by the isotonic function for the bin that contains $x$. To obtain the isotonic function, an optimization problem is solved, which can be expressed as follows:

$$min_{s_1,....,s_n} \sum_{i=1}^{n} (y_i - s_i)^2, \quad (6)$$

subjected to $s_i \leq s_j$ whenever $f(x_i) \leq f(x_j)$, where $f(x_i)$ is the score given by the uncalibrated model, $n$ is the number of samples and $y_i$ is the true labels.

*2) ENsemble-based Confidence caLibration (*ENCL*):* The ENCL combines multiple independent calibration techniques to generate calibrated probabilities. Each technique within the ensemble produces its own calibrated probabilities based on learned parameters. These probabilities are then aggregated by the ensemble method to formulate a final decision. The primary advantage of this approach is its ability to reduce the biases or variances specific to individual calibration methods, often leading to a more generalized and reliable prediction system. ENCL first obtains calibrated probabilities from each method (Spline, Platt, Beta) within the ensemble. ENCL then aggregates these probabilities and applies a voting mechanism where the final probability for each instance is determined by the mode of the combined calibrated probabilities. Here, the mode is defined as the value that appears most frequently among the probabilities calculated by the calibration methods.

Finally, if no mode is present, select the lowest calibrated probability. This final step helps ensure that the ensemble-based calibration method avoids overconfidence in predictions where certainty is low. To clarify further, in information theory, an event with a low probability is less predictable, and the uncertainty associated with that event is high [27], [28]. Entropy, which measures the expected value of uncertainty across all possible events, increases with the presence of low-probability events. This high entropy reflects greater uncertainty, which can lead to reduced confidence in predictions. In a clinical setting where stakes are high, this cautious approach is intended to allow clinicians more room to take necessary precautions to ensure accurate diagnosis and treatment.

*3) Reliability Diagram:* The reliability diagram is a visual tool for assessing the calibration of a model. It involves segmenting predicted probabilities into a fixed number of bins $(N)$, with each bin representing a segment of size $1/N$. These bins are then plotted against the actual outcomes to evaluate the model's prediction accuracy. The accuracy for each bin, denoted as $C_n$, is calculated to understand how well the model's predicted probabilities match the observed outcomes:

$$acc(C_n) = \frac{1}{|C_n|} \sum_{i \in C_n} 1(y_i' = y_i), \quad (7)$$

where $y_i'$ and $y_i$ represent the predicted and true class labels for sample $i$ and $C_n$ is the set of samples whose predicted confidences fall within the interval $I_n = (\frac{n-1}{N}, \frac{n}{N})$, for $n \in 1, 2, ..., N$. The average confidence in the bin $C_n$ is defined as:

$$conf(C_n) = \frac{1}{|C_n|} \sum_{i \in C_n} P_i', \quad (8)$$

where $p_i'$ represents the predicted confidence for each sample $i$. In a model that is ideally calibrated, the accuracy of each bin, the $acc(C_n)$ is expected to be equal to the $conf(C_n)$ for all $n \in 1, 2, ..., N$.

### D. Uncertainty Estimation

Following the approach proposed by Depeweg et al. [29], we utilize predictive entropy as a key metric to assess uncertainty in both calibrated and uncalibrated models. Predictive entropy focuses on the entropy of the predictive posterior distribution, which can be mathematically represented as:

$$H[P(y = k|X)] = - \sum_k P(y = k|X) \log P(y = k|X), \quad (9)$$

where $P(y = k|X)$ is the predicted probability of class $k$ from the DNN. The essence of predictive entropy lies in its ability to gauge the level of uncertainty or ambiguity inherent in model predictions. A higher entropy value signifies greater uncertainty, implying that the model's predictions are dispersed across multiple classes. Conversely, lower entropy indicates a high level of confidence in the prediction, with the model's predictions being more concentrated on a single class. This approach is particularly insightful in our disease diagnosis, where a model's uncertainty can be quantified based on how it allocates probabilities across classes.

Fig. 2. Uncertainty Confusion Matrix. AC represents the number of correctly classified predictions that are certain, while AU represents the number of correct predictions that are incorrectly flagged as uncertain. IU represents the number of incorrect predictions that are correctly flagged as uncertain, while IC represents the number of incorrect predictions that are classified as certain.

## III. EXPERIMENTAL RESULTS

### A. Dataset

We evaluate `ENCL-DNN` on two publicly available COVID-19 cough audio datasets, Coswara and Cambridge. The Coswara dataset, compiled by the Indian Institute of Science (IISc) Bangalore, contains respiratory sounds for disease diagnosis and monitoring [30]. It includes 2746 samples representing seven COVID-19 statuses and focuses on cough sounds. COVID-19 cases were categorized into healthy (1433 individuals) and infected (681 individuals) consisting of positive mild, positive moderate, and positive asymptomatic. The Cambridge dataset [31] from the University of Cambridge includes three participant groups: 141 individuals with COVID-19, 330 without COVID-19 history, and 20 with asthma but no COVID-19 history. Audio samples of cough and breath sounds were collected using an Android app or web platform at a sampling rate of 22050 Hz. This study focuses on categorizing cough sounds into healthy (85 individuals) and infected (141 individuals). For both datasets, we randomly split the training and testing set on the patient level to avoid potential data leakage.

### B. Evaluation Metric

Following [14], [17], we use the area under the receiver operating characteristic (AUROC) to measure the performance of the proposed method on COVID-19 classification. To evaluate the model confidence calibration, we use Log loss, the Brier score, and Expected Calibration Error (ECE). We define four quantitative measures based on the uncertainty confusion matrix in Figure 2 to provide a measure of the uncertainty estimation. These measures are uncertainty accuracy (UAcc), uncertainty sensitivity (USen), uncertainty precision (UPre), and uncertainty specificity (USpe).

### C. Backbone Models

For performance comparison, we have considered several machine-learning algorithms as backbone models, including Logistic Regression (LR), Random Forest (RF), XGBoost, Support Vector Classifier (SVC) with linear (SVC-L), and Radial Basis Function (RBF) kernel (SVC-RBF), and Decision Tree (DT). We also referenced literature such as LSTM [32] and CNN [33].

TABLE I
MAIN RESULTS OF CLASSIFICATION PERFORMANCE (AUROC) ON
COSWARA AND CAMBRIDGE DATASETS.

| Model | Coswara (AUROC) | Cambridge (AUROC) |
|---|---|---|
| LR | 0.762 | 0.672 |
| RF | 0.725 | 0.793 |
| XGBoost | 0.681 | 0.759 |
| SVC-L | 0.764 | 0.789 |
| SVC-RBF | 0.741 | 0.781 |
| DT | 0.570 | 0.797 |
| LSTM [32] | 0.702 | 0.825 |
| CNN [33] | 0.704 | 0.837 |
| ENCL-DNN | **0.834** | **0.854** |

TABLE II
COMPARISON OF CONFIDENCE CALIBRATION USING LOG LOSS, BRIER
LOSS, AUROC, AND ECE ON COSWARA AND CAMBRIDGE DATASETS.

| Dataset | Model | Calibration | Log | Brier | AUROC | ECE |
|---|---|---|---|---|---|---|
| Coswara | LR | - | 0.568 | 0.194 | 0.762 | 0.134 |
| | RF | - | 0.645 | 0.226 | 0.725 | 0.173 |
| | XGBoost | - | 0.978 | 0.260 | 0.681 | 0.238 |
| | SVC-L | - | 0.567 | 0.193 | 0.764 | 0.134 |
| | SVC-RBF | - | 0.567 | 0.195 | 0.741 | 0.118 |
| | DT | - | 7.894 | 0.347 | 0.570 | 0.336 |
| | LSTM [32] | - | 0.587 | 0.199 | 0.702 | 0.108 |
| | CNN [33] | - | 0.137 | 0.256 | 0.704 | 0.227 |
| | DNN | - | 0.558 | 0.187 | 0.834 | 0.178 |
| | | MC Dropout | 0.606 | 0.210 | 0.731 | 0.145 |
| | | Beta | 0.495 | 0.162 | 0.834 | 0.107 |
| | | Isotonic | 0.507 | 0.167 | 0.828 | 0.120 |
| | | Platt | 0.490 | 0.164 | 0.834 | 0.106 |
| | | Spline | 0.486 | 0.161 | 0.834 | 0.104 |
| | | ENCL | **0.486** | **0.159** | **0.834** | **0.089** |
| Cambridge | LR | - | 0.594 | 0.203 | 0.672 | 0.151 |
| | RF | - | 4.237 | 0.178 | 0.793 | 0.131 |
| | XGBoost | - | 0.590 | 0.199 | 0.759 | 0.165 |
| | SVC-L | - | 0.594 | 0.180 | 0.789 | 0.178 |
| | SVC-RBF | - | 0.566 | 0.183 | 0.781 | 0.156 |
| | DT | - | 5.090 | 0.183 | 0.797 | 0.164 |
| | LSTM [32] | - | 0.535 | 0.176 | 0.825 | 0.152 |
| | CNN [33] | - | 0.821 | 0.187 | 0.837 | 0.202 |
| | DNN | - | 0.539 | 0.173 | 0.854 | 0.167 |
| | | MC Dropout | 0.605 | 0.177 | 0.807 | 0.163 |
| | | Beta | 0.724 | 0.202 | 0.854 | 0.204 |
| | | Isotonic | 1.650 | 0.204 | 0.810 | 0.192 |
| | | Platt | 0.525 | 0.174 | 0.854 | 0.133 |
| | | Spline | 0.535 | 0.180 | 0.854 | 0.121 |
| | | ENCL | **0.506** | **0.168** | **0.854** | **0.119** |

### D. Main Results

As shown in Table I, we evaluate the classification performance of `ENCL-DNN` on both the Coswara and Cambridge datasets. Before calibration, for the Coswara dataset, `ENCL-DNN` achieves the highest AUROC score of 0.834, while on the Cambridge dataset, it achieves a score of 0.854. It also suggests that the extracted MFCC features exhibit distinct patterns, effectively differentiating between healthy individuals and COVID-19 patients.

### E. Model Calibration

We then assess the confidence level of `ENCL-DNN` towards its predictions before and after calibration for prediction reliability.

*1) Coswara Dataset:* In Figure 3, we observe that the model confidence consistently improves after implementing the `ENCL`, suggesting that `ENCL-DNN` becomes more confident in its predictions when they are correct, but also acknowledges its uncertainty when a prediction is less certain.

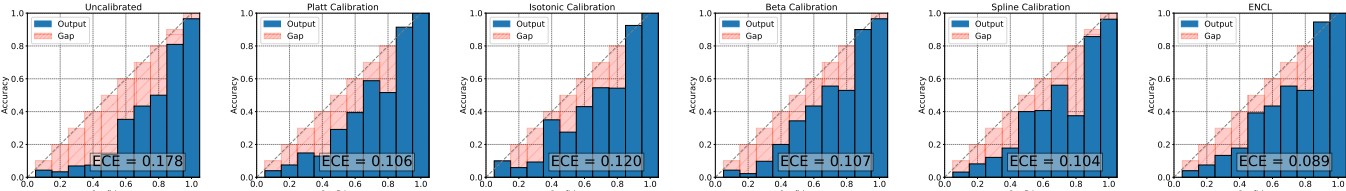

Fig. 3. Reliability Diagram for the Coswara Dataset: ENCL achieves the lowest ECE (0.089) when compared to other calibration methods.

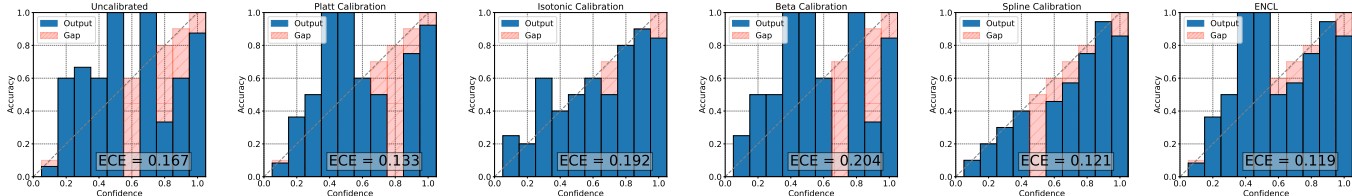

Fig. 4. Reliability Diagram for the Cambridge Dataset: ENCL yields the best ECE (0.119) and appears to be optimally calibrated.

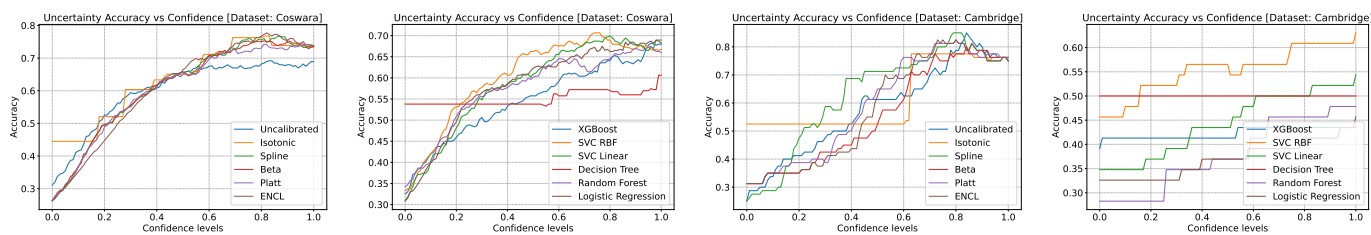

Fig. 5. Uncertainty accuracy as a function of the confidence level. We measure the relationship between model uncertainty and prediction confidence across different methods on the Coswara and Cambridge datasets. The uncertainty accuracy measures the overall accuracy of uncertainty classification. Overall, calibration enhances the model's capability to quantify uncertainty accurately.

Furthermore, we can confirm these results by examining Table II, which illustrates that the ENCL is more effective at improving the model's confidence than other calibration methods. Specifically, ENCL-DNN achieved an ECE score of 0.089, an AUROC of 0.834, a Brier loss of 0.159, and a Log loss of 0.486, all indicating a high confidence level in the model's predictions when ENCL is applied.

*2) Cambridge Dataset:* To demonstrate that ENCL-DNN is effective and generalizable, we have present the reliability diagram of our model on the Cambridge dataset. As shown in Figure 4, we observe that the confidence of our model improves significantly when ENCL is applied. The effectiveness of ENCL can be further verified from Table II, with an ECE score of 0.119, an AUROC of 0.854, a Brier loss of 0.168, and a Log loss of 0.506, which significantly outperforms other calibration methods.

### F. Uncertainty Quantification

In this section, we demonstrate the performance of the models in quantifying uncertainty. We consider the entropy, uncertainty accuracy, and uncertainty matrix. The analysis for the uncertainty confusion matrix was performed using a confidence level of 0.75. This threshold was determined based on the uncertainty accuracy plot in Figure 5. The plot provides information on the model's ability to accurately classify predictions that it was either certain or uncertain about across all confidence levels.

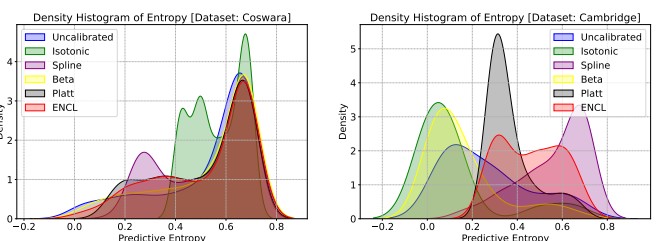

Fig. 6. Density histogram of the entropy on the Coswara and Cambridge datasets. We present the distribution of predictive entropy for various calibration methods (Uncalibrated, Isotonic, Spline, Beta, Platt, ENCL) applied to the Coswara and Cambridge datasets.

*1) Coswara Dataset:* We examine the entropy before and after calibration dissipated in Figure 6, which displays the density histogram of the entropy on the Coswara dataset. We expect the ENCL-DNN to be confident in its predictions when the entropy values are close to 0, but we also want it to be uncertain when it is not confident; that is, when the entropy values are close to 1. We observe that our model captures this behavior well after applying the ENCL. Moreover, Figure 5 illustrates the uncertainty accuracy of the model in identifying accurate and less accurate predictions across all confidence levels. We note that the uncertainty accuracy increases as the confidence level increases when the ENCL is applied. Table III shows the USpe, USen, UPre, and UAcc at a confidence level of 0.75, where we focus on reducing certain incorrect predictions as much as possible for a robust prediction in the

TABLE III
PERFORMANCE COMPARISON OF UNCERTAINTY ESTIMATION ON
COSWARA AND CAMBRIDGE DATASETS.

| Dataset | Model | Calibration | USpe | UPre | USen | UAcc |
|---|---|---|---|---|---|---|
| Coswara | LR | - | 0.492 | 0.768 | 0.749 | 0.670 |
| | RF | - | 0.357 | 0.692 | 0.751 | 0.616 |
| | XGBoost | - | 0.389 | 0.713 | 0.716 | 0.611 |
| | SVC-L | - | 0.528 | 0.780 | 0.755 | 0.685 |
| | SVC-RBF | - | 0.507 | 0.766 | 0.802 | 0.704 |
| | DT | - | 0.206 | 0.613 | 0.807 | 0.572 |
| | LSTM [32] | - | 0.523 | 0.783 | 0.783 | 0.702 |
| | CNN [33] | - | 0.433 | 0.739 | 0.723 | 0.633 |
| | DNN | - | 0.402 | 0.746 | 0.791 | 0.670 |
| | | MC Dropout | 0.420 | 0.716 | 0.745 | 0.636 |
| | | Beta | 0.435 | 0.807 | 0.847 | 0.738 |
| | | Isotonic | 0.417 | 0.809 | **0.887** | **0.763** |
| | | Platt | 0.439 | 0.805 | 0.818 | 0.719 |
| | | Spline | 0.495 | 0.826 | 0.848 | 0.756 |
| | | ENCL | **0.542** | **0.836** | 0.825 | 0.751 |
| Cambridge | LR | - | 0.333 | 0.741 | 0.678 | 0.588 |
| | RF | - | 0.000 | 0.738 | **1.000** | 0.738 |
| | XGBoost | - | 0.350 | 0.787 | 0.800 | 0.688 |
| | SVC-L | - | 0.389 | 0.807 | 0.742 | 0.663 |
| | SVC-RBF | - | 0.381 | 0.797 | 0.864 | 0.738 |
| | DT | - | 0.313 | 0.823 | 0.797 | 0.700 |
| | LSTM [32] | - | 0.245 | 0.566 | 0.838 | 0.566 |
| | CNN [33] | - | 0.330 | 0.552 | 0.702 | 0.531 |
| | DNN | - | 0.700 | 0.878 | 0.717 | 0.713 |
| | | MC Dropout | 0.250 | 0.813 | 0.813 | 0.700 |
| | | Beta | 0.650 | 0.870 | 0.783 | 0.750 |
| | | Isotonic | 0.368 | 0.821 | 0.902 | 0.775 |
| | | Platt | 0.650 | 0.881 | 0.867 | 0.813 |
| | | Spline | 0.650 | 0.879 | 0.850 | 0.800 |
| | | ENCL | **0.700** | **0.897** | 0.867 | **0.825** |

TABLE IV
SUMMARY STATISTICS BETWEEN UNCALIBRATED AND ENCL-DNN ON
COSWARA AND CAMBRIDGE DATASETS.

| Dataset | Property | Metric | Uncalibrated | Calibrated | ↑% |
|---|---|---|---|---|---|
| Coswara | Calibration | Log | 0.558 | 0.486 | 12.903 |
| | | Brier | 0.187 | 0.159 | 14.973 |
| | | ECE | 0.178 | 0.089 | 50.000 |
| | Uncertainty | USpe | 0.402 | 0.542 | 34.826 |
| | | UPre | 0.746 | 0.836 | 12.064 |
| | | USen | 0.791 | 0.825 | 4.298 |
| | | UAcc | 0.670 | 0.751 | 12.090 |
| Cambridge | Calibration | Log | 0.539 | 0.506 | 6.122 |
| | | Brier | 0.173 | 0.168 | 2.890 |
| | | ECE | 0.167 | 0.119 | 28.743 |
| | Uncertainty | USpe | 0.700 | 0.700 | 0 |
| | | UPre | 0.878 | 0.897 | 2.164 |
| | | USen | 0.717 | 0.867 | 20.921 |
| | | UAcc | 0.713 | 0.825 | 15.708 |

real-world scenario.

*2) Cambridge Dataset:* Based on the Cambridge dataset, we analyze the density histogram of the entropy shown in Figure 6. We found that the Platt, Isotonic, and Beta calibration-based methods had a higher density close to the entropy value 0, which implies that more samples fall within this range of entropy. In contrast, ENCL had almost similar densities when entropy values were close to 0 and 1. Moreover, we observed that the model's overall confidence improved slightly when the ENCL was applied, as depicted in Figure 5. Furthermore, Table III showcases the USpe, USen, UPre, and UAcc for each model at a confidence level of 0.75. It is observed that the ENCL exhibits better performance in terms of UPre and USpe, it also demonstrates a marginally better UAcc when contrasted with the other calibration methods considered.

### G. Statistical Analysis

The results of comparing before and after calibration with the ENCL are summarized in Table IV across two main properties: Reliability/Calibration and Uncertainty Estimation. For the Coswara dataset, calibration improved the Log Loss by approximately 12.903%, reduced Brier Loss by about 14.973%, and notably enhanced the ECE by 50%. Uncertainty Estimation metrics, such as UPre, USen, USpe, and UAcc, also showed significant improvements post-calibration, with UPre improving by 12.064%, USen by 4.298%, USpe by 34.824%, and UAcc by 12.090%.

Similarly, for the Cambridge dataset, calibration results in over 6% improvement in Log Loss and 2.890% in Brier Loss. The ECE saw a substantial enhancement of nearly 29%. In terms of Uncertainty Estimation, while UPre slightly improve by about 2%, USen and UAcc increase by 20.921% and 15.708%, respectively. Notably, USpc remained unchanged. These results indicate that ENCL can significantly improve model reliability and the accuracy of uncertainty estimates, which is crucial for developing robust predictive models.

### H. Effect of Activation Functions

We further examine the effect of the activation function in model calibration. Specifically, we examine three cases: no activation function, sigmoid activation function, and softmax activation function. By analyzing the density histogram of each class, as shown in Figure 7, we noticed that when the softmax activation function is applied to obtain the probability distribution, the classes are better separated than when the sigmoid or no activation function is used. Nevertheless, based on the reliability diagram and the entropy displayed in Figure 8, we found that while the softmax activation function exhibited higher density when the entropy values were close to 0, indicating strong confidence in its prediction, it resulted in a poor ECE of 0.250, implying that the model calibration error was poor, and that softmax overestimated the probabilities of its predictions.

### I. Model Interpretation

We utilize the LIME framework [34] to provide a transparent explanation for our model predictions. In Figure 9, we show an accurate prediction with feature interpretations. Upon closer examination of Figure 10, we observe that among the top ten features, approximately seven of them were responsible for this misclassification. To further validate model performance, we apply t-Distributed Stochastic Neighbor Embedding (t-SNE) [35] to visualize the distribution of test samples within the latent space. This technique helps us assess how well the model captures important features and distinguishes between different classes (Figure 11). Upon analysis, we observe that our model before calibration struggles to define an optimal hyperplane that adequately separates the classes. Furthermore, after calibration, we noticed a shift in the distribution, which impacted the separation of the classes. Additionally, when the uncertainty level is incorporated into

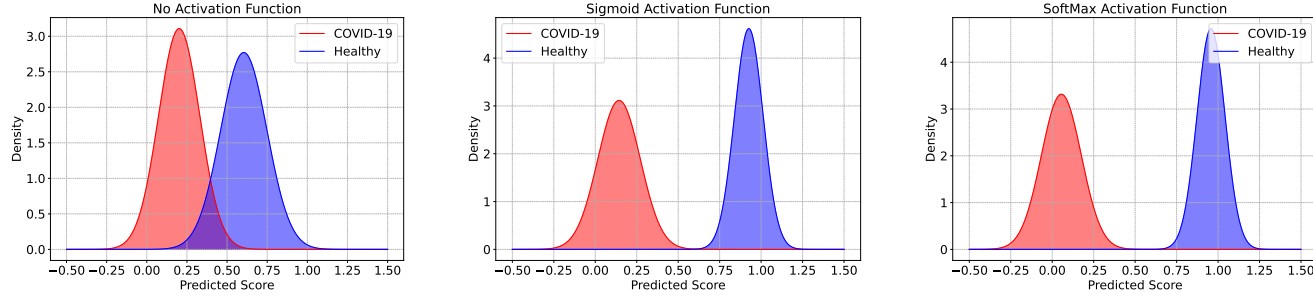

Fig. 7. Density histogram of different activation functions, including Sigmoid and Softmax.

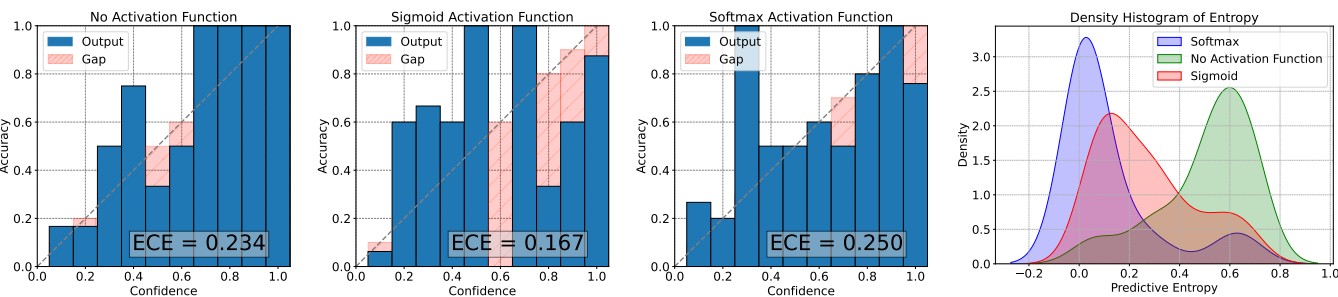

Fig. 8. Reliability diagram and entropy density histogram of different activation functions.

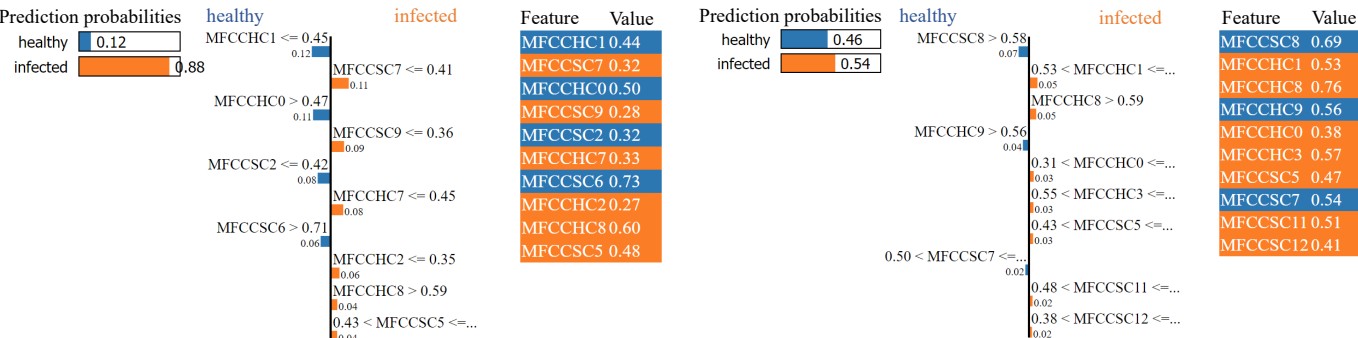

Fig. 9. LIME interpretation of the calibrated model performance on a correctly predicted case study, highlighting the key MFCC features that influence the model's decision, along with the associated prediction probabilities. In this scenario, the model correctly predicts the "infected" class with a probability of 0.88.

the space, we observe that predictions closer to the hyperplane peak at 0.69, further validating that our method works.

## IV. CONCLUSION

The accurate quantification of uncertainty in model predictions is essential for human intervention, especially in the field of medicine. In this study, we propose `ENCL-DNN` for calibrating model confidence and measuring the uncertainty in CDSSs. Our experiments on the Coswara and Cambridge datasets showed that the application of confidence calibration significantly improves the reliability of diagnosis in early COVID-19 detection using audio samples. In addition, we investigated how calibration affects the model's ability to accurately quantify uncertainty in its predictions for a robust diagnosis. This study emphasizes the importance of model calibration, uncertainty estimation, and interpretability to enhance the trustworthiness and reliability of predictions. It also has the

Fig. 10. LIME interpretation of the calibrated model performance on an incorrectly predicted case study. In this scenario, the model incorrectly predicts the "infected" class with a probability of 0.54.

potential to encourage the adoption of mobile healthcare for screening infectious respiratory diseases.

## ACKNOWLEDGEMENT

This research has been supported by a Wallace H. Coulter Distinguished Faculty Fellowship, a Petit Institute Faculty Fellowship, and research funding from Amazon and Microsoft Research to Professor May D. Wang.

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
