# OpenReview forum: "Confidence-Calibrated Clinical Decision Support System for Reliable Respiratory Disease Screening"
_IEEE.org/EMBS/BHI/2024/Conference — IEEE BHI'24_

### Official Review · Reviewer_2Wjx · 2024-08-10
**Confidence-Calibrated Clinical Decision Support System for Reliable Respiratory Disease Screening**

**Overall Rating:** 6
**Confidence:** 3

**Other Quality Metrics:**

a) fair-good
b) fair
c) fair
d) excellent

**Questions For The Authors:**

1. Did the authors test anything besides the Mode when obtaining the calibrated probability?

2. I don't understand why the authors select the lowest calibrated probability when no Mode exists. What if the lowest probability is an outlier?

**Strengths:**

1. Section 3 is extensive and covers every aspect of evaluating this method.

2. While there were a few differences in Log, Brier, AUROC, and ECE between calibration methods, the comparison of uncertainty estimation demonstrates the usefulness of this ensemble method compared to a single calibration method.

**Summary Of The Paper:**

The authors propose an ensemble-based calibration approach to improve respiratory disease (COVID-19) screening based on cough sounds. To do so, they introduce an Ensemble-Based confidence-calibrated deep neural network (ENCL-DNN). Additionally, the authors utilize multiple methods for determining epistemic uncertainty to fine-tune their parameters. From the resulting calibrated probabilities from each of the confidence methods, the authors calculate the mode probability and use this value to calibrate the DNN. compare their findings to several "backbone models," including LR, RF, and several variances of support vector machines.

**Weaknesses:**

1. The methodology section appears somewhat underdeveloped, focusing on defining various confidence calibration methods. This brevity might give the impression that the authors did not invest significant effort into the study's methodological aspects. However, the extensive results and thorough analysis presented suggest otherwise.

2. I would have liked the authors to compare this model to others who have used these datasets for respiratory disease prediction.

3. I would not say ENCL-DNN significantly outperforms the DNN when using a single calibration method in table 2. It outperforms the non-DNN algorithms.

Recommendation) You may want to split up section 3 into 2 or more sections. It is just too long, with too many subsections and subsubsections.

---

> ### Author Rebuttal · Authors · 2024-09-01
>
> Thank you for your insightful review. We appreciate your clear and concise summary of our paper. Your input has been valuable in improving the overall quality of our draft. We will address your comments in more detail below.
>
> > W1: Underdeveloped Methodology
>
> A: We appreciate your input on the methodology section.  Our aim was to recognize the existing post-hoc calibration methods since our proposed method builds from them. Specifically, our approach involves combining these different calibration methods using a mode and adding a constraint to select the lowest value when the highest occurring probability does not exist, which further improves our system. We will further build on the methodology and improve it to capture our contribution accurately.
>
> > W2: Literature Comparison
>
> A: We appreciate the suggestion and recognize its importance. We emphasize that most literature has focused on classification performance rather than calibration and uncertainty levels. Therefore, we will be the first to integrate this. Additionally, we want to point out that most papers focusing on classification have used general backbone models like SVM, RF, etc. This is the reason for evaluating this method in terms of calibration and uncertainty in the paper. Furthermore, we have provided two additional pieces of literature on classification using LSTM and CNN.
>
> | Dataset | Model   | Log  | Bier | ECE   |
> |---------|---------|------|------|-------|
> |         | LSTM | 0.587| 0.199| 0.108 |
> | Coswara | CNN  | 1.137| 0.256| 0.227 |
> |         | ENCL_DNN| 0.486| 0.159| 0.089 |
> |         |         |      |      |       |
> |         | LSTM | 0.535| 0.176| 0.152 |
> | Cambg   | CNN  | 0.821| 0.187| 0.202 |
> |         | ENCL_DNN| 0.506| 0.168| 0.119 |
>
> After observing the table above, we noticed that the calibration error increases significantly in the CNN model compared to the LSTM model. This discrepancy can be attributed to the fact that the literature emphasizes the use of SoftMax as the activation function, which further confirms our conclusion in section III-H that SoftMax tends to overestimate the probability. We have included these additional results in the revised manuscript.
>
> > W3: ENCL-DNN Performance
>
> A: We appreciate your observation regarding the performance of ENCL-DNN. While we agree that ENCL-DNN outperforms other non-DNN algorithms, we also emphasize that it delivers better performance compared to uncalibrated DNNs, as evidenced by the results in Table IV. Our primary objective with ENCL-DNN was not just to surpass DNNs in performance but to enhance overall robustness and improve uncertainty estimation. By integrating the strengths of various calibration methods, ENCL-DNN effectively balances their advantages, resulting in better calibration performance while doing its best to mitigate their individual limitations.
>
> > Q1: Other methods considered?
>
> A: Yes, we did explore other ensemble methods specifically, voting methods besides the Mode + our constraints to obtain the calibrated probability. We have provided a comprehensive result of this approach.
>
> |Dataset | Model         |  Log  |  Bier |   ECE  |
> |--------|---------------|-------|-------|--------|
> |        | Average Voting| 0.488 | 0.162 | 0.105  |
> |Coswara | Max Voting    | 0.492 | 0.161 | 0.104  |
> |        | ENCL_DNN      | 0.486 | 0.159 | 0.089  |
> |        | Average Voting| 0.507 | 0.168 | 0.145  |
> |Cambg   | Max Voting    | 0.625 | 0.183 | 0.136  |
> |        | ENCL_DNN      | 0.506 | 0.168 | 0.119  |
>
> However, from the result above, we found that our method performed best and provided the most consistent results across different scenarios.
>
> > Q2: Why the constraint?
>
> Thank you for raising this important point. We selected the lowest probability as a conservative approach to avoid overestimating confidence when there isn't a clear highest occurring value. To clarify further, in information theory, when an event has a low probability, it is less predictable, and the uncertainty associated with that event is higher. Entropy, which measures the expected value of uncertainty across all possible events, increases in the presence of low probability events. This higher entropy reflects greater uncertainty, which in turn leads to less confidence in predictions. This cautious approach is intended to allow clinicians more room to take necessary precautions in a clinical setting.
>
> Moreover, if the lowest probability is identified as an outlier, we anticipate that this could further elevate uncertainty, providing clinicians with an additional prompt to exercise caution. We have provided a more detailed explanation of our rationale in the revised manuscript.
>
> >References
> -  Shannon, C. E. (1948). A mathematical theory of communication
> - Cover, T. M. (1999). Elements of information theory. John Wiley & Sons.
>
> Thank you for your review! We hope our responses can address your concerns. Please let us know if you have any questions, and we are happy to discuss further.

---

### Official Review · Reviewer_WM4H · 2024-08-10
**Confidence-Calibrated Clinical Decision Support System for Reliable Respiratory Disease Screening**

**Overall Rating:** 8
**Confidence:** 4

**Other Quality Metrics:**

(a) Clarity of writing: good
 (b) Clinical Significance: good
 (c) Methodological Novelty: good
 (d) Experiments and Results: good

**Questions For The Authors:**

1. What is the motivation and application of this study?

**Strengths:**

This paper has good clinical significance. The description is clear, and the experiments are substantial.

**Summary Of The Paper:**

This paper propose ENCL-DNN for calubrating model confidence and measuring the uncertainty in CDSSs. It also investigates how calibration affects the model's ability to accurately quantify uncertainty in its prediction for a robust diagnosis.

**Weaknesses:**

There are some clerical errors in the text

---

> ### Author Rebuttal · Authors · 2024-09-01
>
> We sincerely appreciate your thoughtful review and are particularly grateful for your recognition of our paper's clinical significance, clear description, and substantial experiments. Your feedback has been invaluable in helping us improve our manuscript.
>
> > W1: There are some clerical errors in the text.
>
> A: We thank you for bringing the clerical errors to our attention. We will conduct a thorough review and correct these errors to improve the overall quality and readability of the updated manuscript.
>
> > Q1: What is the motivation and application of this study?
>
> A: Thank you for your insightful question regarding the motivation and application of our study.
>
> **Technical Motivation**: While the design of an efficient neural network and the optimization of hyper-parameters can enhance model performance, it does not guarantee proper calibration. As emphasized in our paper, poorly calibrated models produce predicted probabilities that do not accurately reflect the true likelihood of correctness [1]. Several factors, such as the depth and width of the model, batch normalization, and weight decay, can influence model calibration, and these were carefully considered in our study.
>
> **Clinical Motivation**: Given the critical nature of accurate predictions in high-stakes medical environments, ensuring proper calibration becomes even more crucial. The primary motivation for this study is to address the increasing demand for reliable and robust Clinical Decision Support Systems (CDSSs) that can provide accurate and well-calibrated predictions. Our goal is to develop a model that is confident in its correct predictions while demonstrating less confidence when the predictions are incorrect. This highlights the importance of clearly communicating the uncertainty levels in our model's outputs.
>
> **Application**: The application of our proposed ENCL-DNN model extends to any CDSSs, beyond just the audio data used as a proof of concept, where understanding the confidence and uncertainty of predictions is critical. This includes areas such as complex disease diagnosis, treatment planning, and personalized medicine. By focusing on calibration, we aim to improve the reliability of CDSSs, ensuring that they can better support clinicians in making informed decisions, ultimately leading to improved patient outcomes.
>
> [1] Chuan Guo, Geoff Pleiss, Yu Sun, and Kilian Q Weinberger. 2017. On calibration of modern neural networks. In International conference on machine learning. PMLR
>
> Thank you for your review! We hope our responses can address your concerns. Please let us know if you have any questions, and we are happy to discuss further.

---

### Decision · Program_Chairs · 2024-09-23

Accept